# The Impact of COVID−19 on Female Sexual Health

**DOI:** 10.3390/ijerph17197152

**Published:** 2020-09-30

**Authors:** Anna Fuchs, Aleksandra Matonóg, Joanna Pilarska, Paulina Sieradzka, Mateusz Szul, Bartosz Czuba, Agnieszka Drosdzol-Cop

**Affiliations:** 1Chair and Department of Gynecology, Obstetrics and Oncological Gynecology, Medical University of Silesia in Katowice, Markiefki 87, 40-211 Katowice, Poland; aleksandra.matonog@gmail.com (A.M.); yoasiapil26@gmail.com (J.P.); pausieradzka@gmail.com (P.S.); mszul100@gmail.com (M.S.); cor111@poczta.onet.pl (A.D.-C.); 2Chair of Woman’s Health, Medical University of Silesia in Katowice, Medyków 12, 40-752 Katowice, Poland; bartosz.czuba@sonomed.net

**Keywords:** COVID−19 virus, coronavirus disease 2019 (COVID−19), SARS-CoV−2 pandemic, sexuality, sexual dysfunction, desire

## Abstract

Introduction: Coronavirus disease (COVID−19), announced as a pandemic by the World Health Organization, recently has dominated people’s lifestyle. The impact of COVID−19 seems to be relevant to the sexual health as well. Methods: This prospective study was conducted on two occasions involving 764 female patients between March and April 2020—before and during the time of social quarantine. The sexual function was assessed using the Polish version of the Female Sexual Function Index (FSFI). Every patient filled out the survey concerning socio-demographic characteristics as well as the influence of SARS-CoV−2 pandemic on their lives. Results: The overall FSFI score before the pandemic was 30.1 ± 4.4 and changed to 25.8 ± 9.7 during it. Scores of every domain: desire, arousal, lubrication, orgasm, satisfaction and pain decreased as well (*p* < 0.001). There was statistically significant association between the workplace and the change of FSFI scores before and during COVID−19 pandemic (*p* < 0.01). We noticed the biggest decrease in FSFI score in the group of women who did not work at all (5.2 ± 9.9). Religion had a statistically important impact on level of anxiety (*p* < 0.01). Conclusion: The main finding of our study was the influence of COVID−19 pandemic on the quality of sexual lifestyle and frequency of intercourse among Polish women.

## 1. Introduction

Sexuality is an important part of women’s life and one of the most important one’s responsible for mental health. It has a huge impact on maintaining interpersonal communication and determines human well-being. Among the benefits of a satisfying sex life mentioned are experiencing pleasure, relieving sexual tension and expressing emotional closeness. Human sexuality is controlled by many internal and external factors such as anatomy, hormones and emotions [1,2].

The Female Sexual Function Index (FSFI) is one of the most frequently used scales to measure the sexual functioning, considered as a “gold standard”. It assesses various aspects of sexuality. Conducting the survey twice gives a possibility to check the impact of some factors on sexual life. Hence, answers are given before and after the action of the factor whose impact we want to examine. Thanks to medical advances, it seems crucial to recognize impairments in female sexual life, which when untreated are associated with higher risk of depression, anxiety and lower overall quality of life [3].

### Coronavirus Disease (COVID−19)

Previously unknown or underestimated infectious diseases, especially those which spread very quickly, may affect the human psyche. Coronavirus disease (COVID−19), announced as a pandemic by the World Health Organization, lately has dominated people’s lifestyle. The isolation and monotony of everyday life make time go by significantly differently. Moreover, concomitant stress and anxiety might lead to mood swings and depression or decrease sexual desire. However, due to remote working, many couples spend more time together than usual. Therefore, the impact of COVID−19 seems to be relevant to sexual health.

Severe acute respiratory syndrome coronavirus 2 (SARS-CoV−2) is a novel and unknown pathogen isolated in 2019 in Wuhan, Hubei, China. However, the first virus of *Coronaviridae* family had been isolated in humans in 1962. Previous outbreaks of coronavirus infections appeared in 2002−2003, when the world experienced Severe Acute Respiratory Syndrome (SARS) caused by SARS-CoV, with 744 deaths and 8096 total cases, and in 2011, with the named Middle East Respiratory Syndrome (MERS) caused by MERS-CoV, with 858 deaths and 2494 total cases. All of the human coronaviruses cause upper and lower respiratory tract infections. Despite the fact that medicine has already faced the problem of SARS-CoV and MERS-CoV, belonging to the same family, SARS-CoV−2 shows only ~79% similarity to SARS-CoV and ~50% similarity to MERS-CoV, whereas it seems to be closely related to two bat-derived SARS-like coronaviruses, namely bat-SL-CoVZC45 and bat-SL-CoVZXC21 (about 88−89% similarity) [4,5].

The outbreak of COVID−19 progressed very quickly to pandemic. The first case of coronavirus (SARS-CoV−2) in Poland was reported on 4 March (Ministry of Health of the Republic of Poland report). As a result of a rapid increase in the number of cases, on 10–12 March, the Polish government decided to reduce personal contact by closing a lot of public places such as schools, universities, cinemas and restaurants. People who cross the Polish border or have had contact with contagious persons are obligated to be quarantined for 14 days. Lockdowns had forced many companies to close; thus, a lot of people have suddenly lost their job and income. Unemployment problems and uncertain future are surely connected with additional stress and fear.

The incubation period of SARS-CoV−2 is not precisely defined, and clinical manifestations include fever, cough, dyspnea, myalgia, headache or diarrhea. At the same time, new symptoms are being added to an alarming list. Furthermore, rapid transmission and ability to get infected before the symptomatic phase and likewise a possibility of infection being completely asymptomatic made meeting with other people more stressful, because of the uncertainty about who is infected. It applies especially to those, who have other diseases. Furthermore, it is not known yet what are the far-reaching complications of COVID−19, whereas there is evidence of permanent changes in the lungs, which increases the anxiety level [5,6,7].

Likewise, access to health care centers might be now difficult. This problem may also be compounded by the patients’ fear of interpersonal contact and the higher possibility of infection in public places. Most of the COVID−19 studies conducted until today have focused on assessing the impact of the pandemic on physical health. This prompted us to reflect on the mental and sexual consequences that we consider to be equally important. Guanjian Li et al. have already shown the direct impact of COVID−19 on sexual health, which manifested as decreased sexual desire and frequency of sexual intercourse due to COVID−19. Moreover, Pedrozo-Pupo et al. studied the impact of the pandemic on stress and noted that 15% of participants have higher stress levels due to COVID−19 pandemic. None of the studies conducted until today have had the same aim as our study, which is the association between COVID−19 pandemic, associated anxiety or stress and impact on female sexual health, to consider if women are in need of counseling with a sexologist [8,9].

## 2. Materials and Methods

This prospective, observational, non-interventional study was carried out at the Department of Pregnancy Pathology, Department of Woman’s Health, School of Health Sciences in Katowice, Medical University of Silesia, Poland, between March and April 2020.

Women sexually active, in childbearing age were enrolled in our study. Exclusion criteria were age under eighteen years old, established diagnosis of COVID−19, mental illness including depression or personality disorders and using medicine that reduces libido for the previous three months. These inclusions and exclusions resulted in 764 patients for the present analyses.

Female Sexual Function Index (FSFI) questionnaire is a self-administered screening survey for evaluating the presence of sexual dysfunction, and it contains 19 items referring to 6 different aspects of sex life: desire, arousal, lubrication, orgasm, satisfaction and pain. Total minimum score in FSFI scale is 2 and maximum is 36. A score of 26 or lower indicates a female sexual disfunction. The assessment refers to the previous 4 weeks. It was conducted twice, by email. First surveys had been gathered at the beginning of March, before the first case of COVID−19 in Poland, while many European countries had been announcing numerous cases and when there were no government restrictions in Poland yet. Hence, the answers referred to the four weeks of February. Lockdown in Poland began on March 13 so that the second questionnaire was conducted in second half of April and referred to the time of social quarantine [10,11].

Information about age, habitation, education, marital status, housemates, living conditions, pregnancy and faith was collected during the first survey. Questions in the second questionnaire concerned whether the pandemic had an influence on deterioration of material standing, workplace change and the frequency of sexual intercourse. If this number increased or decreased, compared to the first survey, we asked about the reasons. Questions about SARS-CoV−2 were about mandatory quarantine and infections in womens’ closest environment. Moreover, both questionnaires contained questions about levels of stress and anxiety. We evaluated the level of these parameters within ten-point scales which were very understandable and transparent to women, to assess their own general sensation.

The university Ethics Committee waived the requirement for informed consent due to the non-interventional nature of the study (PCN/0022/KB1/108/20).

All data analyses were conducted using Dell, StatSoft Polska, Statistica version 13.0 PL, 2019 software and a *p*-value < 0.05 was considered as significant. Analyses of dependent variables before and during quarantine were evaluated by the Wilcoxon’s rank test. For the sake of quantitative variables, comparison Chiˆ2 was utilized. Kruskal–Wallis test was implemented for independent variables, and further investigations were accomplished by the U Mann–Whitney test. Data are presented as a mean ± SE (Standard Error).

## 3. Results

Examined women were in the 18−40 age bracket, and the average age of respondents equaled 25.1 ± 4.3. The majority of women had higher (74.2%) or secondary (24.5%) education and lived in cities above 500,000 residents (33.1%). Most of the respondents were in informal relationships (68%), 24.8% were married, and only 7.2% of them were declared single. Almost three quarters of women have never been pregnant before (72,4%). The most common religion was Catholicism (63.5%). Table 1 presents general characteristics of the studied group (Table 1)

Overall FSFI score before the pandemic was 30.1 ± 4.4 and changed to 25.8 ± 9.7 during it. In addition, scores of every domain: desire, arousal, lubrication, orgasm, satisfaction, and pain decreased as well (*p* < 0.001). Desire decreased from 4.5 ± 1.0 to 4.2 ± 1.3, arousal 5.1 ± 0.9 to 4.1 ± 2.0, lubrication 5.4 ± 0.8 to 4.5 ± 2.1, orgasm 4.8 ± 1.3 to 3.9 ± 2.1, satisfaction 5.2 ± 1.0 to 4.7 ± 1.4 and pain 5.1 ± 1.1 to 4.3 ± 2.1. Table 2 shows the mean overall FSFI and several domains scores prior and during pandemic.

The number of women with sexual dysfunction (overall FSFI score 26 or below) before the pandemic was 15.3% and increased to 34.3% during total lockdown; thus, association was statistically significant (*p* < 0.001). Moreover, during the pandemic, the frequency of sexual intercourse declined compared to the period before, and this association is demonstrated in Figure 1. The majority of women declared that the reason might be associated with isolation from the partner (41.5%), 39.3% felt lack of desire caused by stress, and 16% chose misunderstandings with their partners as a cause. Finally, only 3.2% women feared that SARS-CoV−2 could be transmitted during sexual contact.

The statistical analysis of the FSFI score reveals that the mean overall value was 27.9 ± 6.0. No significant association was found (*p* > 0.05) between age, place of residence, marital stage, education, number of pregnancies, religion and mandatory or self-quarantine. The statistically important impact on mean overall FSFI had education and living conditions. Well-educated women had the highest score (28.2 ± 5.9), whereas ones with secondary and primary educations had in sequence 27.2 ± 6.4 and 27.6 ± 5.9 score. Counting the living conditions, women with very good living conditions had the highest mean overall FSFI score (28.8 ± 6.0), in comparison to ones with good ones (27.6 ± 5.9) and average ones (26.9 ± 6.3). Of all, 35% mentioned the deterioration of material standing; however, it was not statistically important to their sexuality.

Furthermore, during the pandemic, overall stress and anxiety levels increased. Those changes are statistically important and are shown in Figure 2.

The results of our study demonstrated that there was statistically significant association between the workplace and change of FSFI score before and during COVID−19 pandemic (*p* < 0.01). We noticed a bigger decrease in FSFI score in the group of women who did not work at all (5.2 ± 9.9) than in those who worked remotely (4.1 ± 8.5) and the smallest change of score in women working outside the house (1.4 ± 5.4). Moreover, the marital stage and type of housemate were statistically significant (*p* < 0.001) for the decrease in FSFI score during COVID−19. Those associations are revealed in Figure 3 and Figure 4. The biggest difference in scores was observed in the group of single women (6.9 ± 10.5). The score of those in informal relationships decreased by 4.7 ± 9.3 and by 2.5 ± 7.5 in the group of married women. We noticed the greatest decrease of FSFI points in women who live with their parents (9.3 ± 11.4). A smaller change was observed in the group of women who lived alone (6.8 ± 10.3) than in those who live with both partner and child (2.5 ± 7.5) or only with a partner (1.4 ± 5.6).

Additionally, religion had a statistically important impact on level of anxiety (*p* < 0.01); the biggest increase in anxiety was noticed in the group of Catholic women compared to unbeliever women. During the lockdown, the level in Catholic women increased from 3.5 ± 2.1 to 6 ± 2.5 (differential 2.5 ± 2.4) and in atheists from 4 ± 2.2 before the pandemic to 5.8 ± 2.4 (differential 1.8 ± 2.3).

## 4. Discussion

The overall point result of FSFI survey during the pandemic significantly decreased, which undeniably indicates the influence of the pandemic on the deterioration of the quality of sex life among Polish women. It was also confirmed by a reduction in everyone of the 6 aspects of sex life included in the questionnaire. Moreover, the frequency of sexual intercourse decreased. Another study also examined the effect of COVID−19 pandemic on female sexual behavior in women in Turkey. The results partly corresponded to ours. Yuksel and Ozgor reported that the pandemic is associated with decreased quality of sexual life, while sexual desire and frequency of intercourse increased during lockdown. Furthermore, they noted that the pandemic had an influence on the decreased desire for pregnancy, decreased female contraception and increased menstrual disorders. Micelli et al. concluded that the majority (66,4%) of Italian people who did not experience desire for parenting before and also during the COVID−19 pandemic, reported no reduction in frequency of sexual intercourse, with no significant differences among genders [12,13].

Furthermore, COVID−19 pandemic contributed to the huge increase in stress and anxiety levels but also to the significant decrease in the amount of intercourse, which was mostly caused by isolation and lack of desire as a consequence of stress. Hall et al. examined 992 young women (age 18−20) with stress symptoms and their impact on sexual health. This study reported a positive association of depression symptoms with frequency of sexual intercourse. In comparison, another study conducted by Liu et al. examined reproductive health of women after the massive earthquake in 2010 in Asia. It turned out that a major disaster leads to widespread sexual disfunction and lower satisfaction of sexual life. That may signify that once the problem concerns everyone around in a similar way and might have consequences for the future, its impact on sexual aspects is rather negative. Moreover, it is probable that age has a major role in coping with stress [14,15].

On the other hand, the lower frequency of sexual contacts might be connected with women’s stress about a possible pregnancy. Fortunately, studies have showed that mortality of pregnant women with COVID−19 is significantly lower (0%) compared to Severe Acute Respiratory Syndrome (SARS) (18%) and Middle East Respiratory Syndrome (MERS) (25%). In SARS and MERS, the most common causes of death were progressive respiratory failure and severe sepsis. Micelli et al. (2020) evaluated the impact of COVID−19 on the desire for parenthood in Italian people of reproductive age with stable relationships. More than one-third of couples, who were planning to have a child, decided to interrupt the pursuit at the time of the pandemic. Among reasons, researchers named mainly the fear of economic instability and lack of knowledge of pregnancy outcomes related to infection. Moreover, the changes in cardiorespiratory and immune systems occurring during the pregnancy might increase the susceptibility to serious infections. Interestingly, the study has also revealed that about 12% of people (mostly women) started to desire parenthood at the time of the quarantine [13,16].

To our knowledge, this study is the first one to establish an association between the quality of sexual life and workplace and marital stage during COVID−19 pandemic. Women not working during the pandemic showed the biggest decrease in FSFI score compared to those working from home. The smallest difference was noted in those working outside the home. This may mean that in the group of non-working women, the causes of FSFI decrease are lack of activity and fatigue due to the everyday routine. A study conducted by Lee et al. somehow confirms our reports. They demonstrated that family income can affect sex life satisfaction; therefore, women who work on a normal basis are not only exposed to less stress but also have constant salary, which may lead to a more satisfying sex life compared to women who do not work at all. Likewise, the biggest decrease was reported among single women, a smaller one in those who are in a relationship and the smallest in marriages. Although normally relationship satisfaction plays the main role in sex life quality, we suggest that the lockdown limits the opportunities to meet new people and make new relationships [17,18].

The important aspect of female sexuality and one of the most common disorders in Female sexual dysfunction (FSD) is vaginal lubrication, and it significantly decreased during the COVID−19 pandemic. Lubrication disorders lead to many problems, such as dyspareunia, orgasm dysfunction, vagina irritation or increased risk of vaginitis, so gynecologists, during the examination, should pay more attention to these aspects. Another study conducted by Yuksel and Ozgor indicated that vaginal lubrication increased during the lockdown, but this difference was not statistically significant [12,19].

The biggest decrease in FSFI score was noted among women living with their parents, then those living alone and finally those who live not only with their partner but also with a child. The smallest differences were noticed in the case of women who live only with their partner. However, living with parents seems to have an obvious impact on intimacy, and living alone reduces the ability of meeting people, an interesting fact is the bigger change in FSFI score in families with children compared to those who are childless. Stavdal et al. examined intimacy and sex life between parents after the first childbirth. They emphasized the fact that in many cases babies take all of the attention and eventually parents have less intimacy and time for each other. In our opinion, additionally anxiety about a child’s health during pandemic plays the main role in deterioration of relations between the lovers [20].

The weakness of the research is that the investigation protocol did not include questions to assess marital satisfaction and conflicts. We are aware that there might be countless factors, which can coincide in time with the lockdown and may have an impact on sexual satisfaction. However, to exclude those groups, we would like to have an opportunity to take a detailed history with our patients in a subsequent study. This paper is a pilot study, so more research needs to be done.

## 5. Conclusions

The main finding of our study was the important influence of COVID−19 pandemic on the quality of sexual life and frequency of intercourse among Polish women. In addition, it eventually affected the level of stress and anxiety. The attention of doctors is now mainly focused on infectious diseases and their symptoms, but in this unusual situation we must not forget that a pandemic affects many different aspects of human health, including sex life. Therefore, if we want to avoid long-term effects and improve quality of life, we should pay attention to sex satisfaction at every medical visit. Additionally, our study showed that certain relationships between partners deteriorated, and the levels of stress and anxiety increased; thus, the role of every doctor is getting this information in anamnesis and if applicable suggesting psychotherapy. Moreover, we should consider how to facilitate contact with sexologists in this difficult time of limited interpersonal contacts, for which telemedicine can help us a lot [21].

## Figures and Tables

**Figure 1 ijerph-17-07152-f001:**
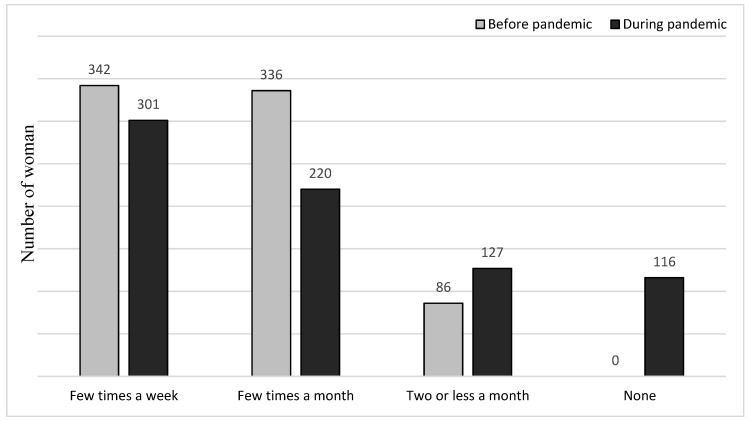
Frequency of sexual intercourse before and after pandemic.

**Figure 2 ijerph-17-07152-f002:**
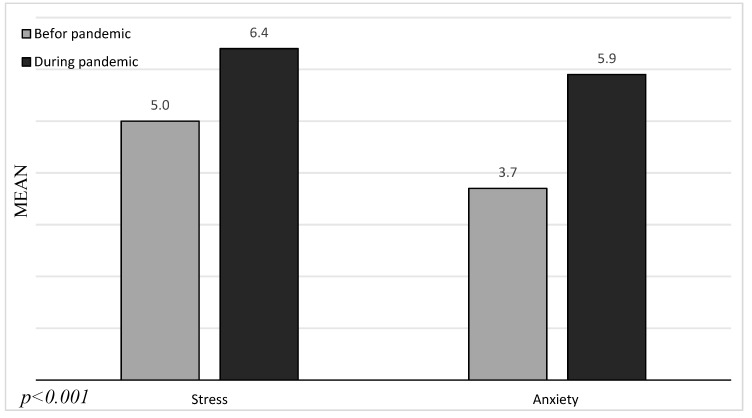
Mean overall stress and anxiety scores before and during the pandemic.

**Figure 3 ijerph-17-07152-f003:**
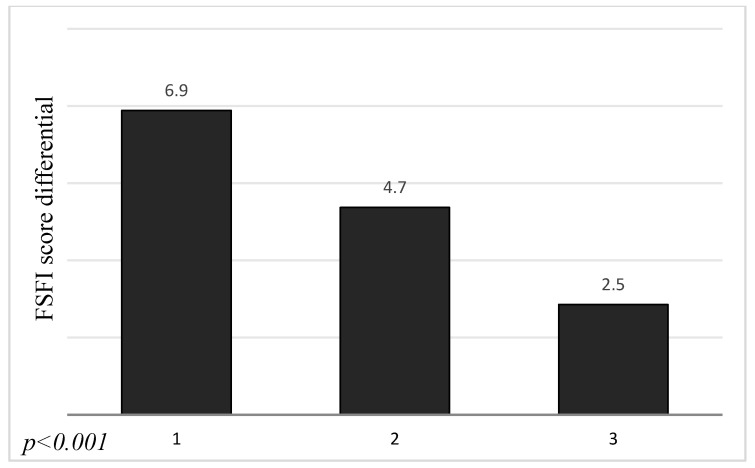
Association between FSFI score differential and marital stage.

**Figure 4 ijerph-17-07152-f004:**
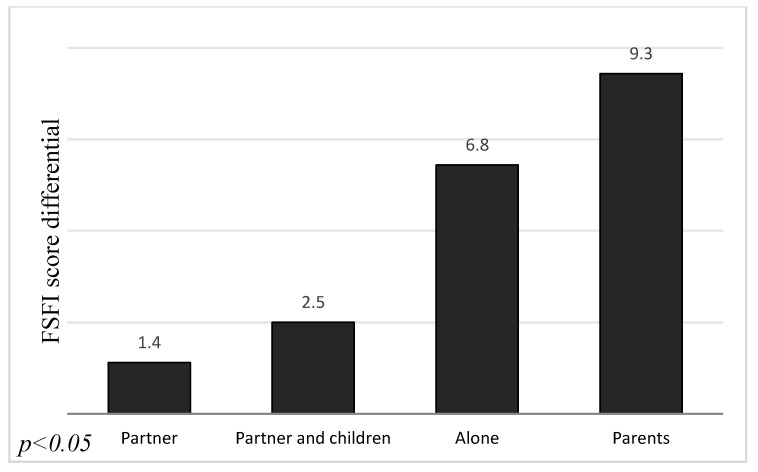
Association between FSFI score differential and type of a housemate.

**Table 1 ijerph-17-07152-t001:** Demographic characteristics of the study participants.

Characteristics	
**Age [years]**	
Average	25.1 ± 4.3
**Marital Status**	
Informal relationship	519 (68%)
Married	190 (24.8%)
Single	55 (7.2%)
**Education**	
Primary	10 (1.3%)
Secondary	187 (24.5%)
Higher	567 (74.2%)
**Place of residence**	
City above 500,000 residents	253 (33.1%)
City 250,000−500,000 residents	138 (18.1%)
City 50,000–250,000 residents	131 (17.1%)
Town below 50,000 residents	116 (15.2%)
Village	126 (16.5%)
**Living conditions**	
Very good	294 (38.4%)
Good	351 (46%)
Average	119 (15.6%)
**Pregnancy**	
Never	553 (72.4%)
Once	100 (13.1%)
More than once	88 (11.5%)
Actually	23 (3%)
**Religion**	
Catholicism	485 (63.5%)
Atheism	229 (30%)
Other	50 (6.5%)

**Table 2 ijerph-17-07152-t002:** Female Sexual Function Index (FSFI) before and during pandemic.

Variables	Before Pandemic	During Pandemic	*p*
Desire	4.5 ± 1.0	4.2 ± 1.3	<0.001
Arousal	5.1 ± 0.9	4.1 ± 2.0	<0.001
Lubrication	5.4 ± 0.8	4.5 ± 2.1	<0.001
Orgasm	4.8 ± 1.3	3.9 ± 2.1	<0.001
Satisfaction	5.2 ± 1.0	4.7 ± 1.4	<0.001
Pain	5.1 ± 1.1	4.3 ± 2.1	<0.001
Total FSFI	30.1 ± 4.4	25.8 ± 9.7	<0.001

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
