# Peer review of "The Impact of COVID−19 on Female Sexual Health"

_ijerph, 2020, doi:10.3390/ijerph17197152_

Round 1

Reviewer 1 Report

Figure 3 and 4 no plots could bee seen on these figures.

Author Response

Dear Reviewer,

Thank you for your positive opinion and suggestions. We sincerely apologize for technical difficulties in opening - Figure 3 and 4. Of course, we improved it. We strongly believe that we have fulfilled your expectations because we put a lot of work into this manuscript.

Yours sincerely,

The Authors

Reviewer 2 Report

In the introduction, the authors should include information from the scientific literature on "... the association between COVID 19 pandemic, associated anxiety or stress, and female sexual health, to consider if women are in need of sexology telemedicine". The text they presented is missing from this information.

The sample should have excluded women with a history of sexual dysfunction. This was not done, which biased the composition of the sample, especially since the investigation protocol did not include means to overcome this difficulty.

The mandatory quarantine associated with COVID, in some couples, caused relational difficulties.

  It is strange that the investigation protocol did not include questions to assess marital satisfaction and conflicts.

These weaknesses in the elaboration of the paper should have been assumed in the item "discussion", which unfortunately did not happen.

Author Response

Dear Reviewer,

Thank You for Your comments and suggestions. Let me please show you step-by-step the changes that we have made:

 #1 In the introduction, the authors should include information from the scientific literature on "... the association between COVID 19 pandemic, associated anxiety or stress, and female sexual health, to consider if women are in need of sexology telemedicine". The text they presented is missing from this information.

Thank you very much for your valuable suggestion. We added information in 93-101 lines according to studies conducted till today.

#2 The sample should have excluded women with a history of sexual dysfunction. This was not done, which biased the composition of the sample, especially since the investigation protocol did not include means to overcome this difficulty. and #3 The mandatory quarantine associated with COVID, in some couples, caused relational difficulties.

 Dear Reviewer,thank you for this remark, we find it very helpful. Obviously we agree with the fact that history of sexual dysfunction impacts sexual satisfaction, but we believe that there are countless factors which are not connected with the SARS-CoV2 virus, that can strongly impact women's sexual satisfaction as well.

In our study we excluded patients with mental illness including depression or personality disorders and using medicine reducing libido for last three months, however, indeed we did not ask about sexual dysfunction in the past, what we obviously will do in our subsequent manuscripts.

What is more, we agree with the fact that mandatory quarantine may result in more frequent relational difficulties. Besides, remote work, more time spent together and other factors connected to pandemic might also contribute to these types of problems, so it is hard to assume that it concerns only the mandatory quarantine to distinguish that group. More frequent relational conflicts are undeniably one of the effects of pandemic in some relations, and those have an influence on sexual satisfaction, for this reason we wanted to study the general impact of pandemic and its effects on sexual satisfaction.

This is our pilot study and we will certainly try to introduce suggested by Reviewer changes of the investigation protocol to the future manuscripts related to COVID 19 pandemic.

#4 It is strange that the investigation protocol did not include questions to assess marital satisfaction and conflicts.

As mentioned, this is our pilot study and we will certainly try to introduce suggested by Reviewer changes of the investigation protocol to the future manuscripts related to COVID 19 pandemic.

If there are any intermarital conflicts or low relationship satisfaction, they affect a woman's sexuality both before and during the pandemic. Therefore, these are factors that constantly accompany a woman and should not distort the results of work that focuses on the impact of the pandemic on sexuality.
We try our best to find the best research mean to assess the sexual dysfunction,in order to help clinicians remember not only about the issue of infection but also about the long-term effects of the COVID 19  pandemic on the quality of life.

#5 These weaknesses in the elaboration of the paper should have been assumed in the item "discussion", which unfortunately did not happen.

Thank you very much for your valuable remark. We improved the discussion according to your comments, obviously adding the information about the weaknesses of the study. We are aware that there might be countless factors, which can coincide in time with the lockdown and may have an impact on sexual satisfaction. However, to exclude those groups we would like to have an opportunity to take a detailed history with patients face-to-face, which is very difficult to do during the lockdown. Our paper is a pilot study on this subject, so more researches are needed to be done.

Thank you once again for all your suggestions. We strongly believe that we have fulfilled your expectations because we put a lot of work into this manuscript.

Yours sincerely,

The Authors

Reviewer 3 Report

In their paper titled " The impact of COVID-19 on female sexual health", Fuchs et al. describe that the critical role of miR-142-3p on the pathogenesis of preeclampsia through the inhibition of trophoblast proliferation and invasion. The authors investigated the change of sexual activity before and after the pandemic of covid-19. The authors assessed the tendency of the quality of sexual lifestyle and frequency of intercourse. This is an interesting manuscript and valuable for many clinicians. Also, the aim is sufficiently original. However, the overall quality of the manuscript is impaired by some problems including the quality of the data and the poor methods. Especially, there is a critical problem in the content of discussion. Furthermore, the overall quality of the manuscript would be improved by a thorough revision of the overall course. It is thought that further proofreading is needed to improve the quality of the text. I have provided specific suggestions below.

Major concerns:

#1. Introduction: There is no mention in the introduction why the authors focused on the sexual lifestyle and covid-19.

#2. Methods: They commented on some data in this manuscript; however, if they would like to describe something about the results, they first need to perform multivariate analysis.

#3. Figure 3 and 4: please recheck these.

#4. Discussion: The sentences are lengthy and include unnecessary sentences.

#5. P8, L212-226: The sentences are not needed.

#6. Discussion: I cannot understand what the authors like to conclude from the data.

Author Response

Dear Reviewer,

Thank You for Your comments and suggestions. Let me please show you step-by-step the changes that we have made:

#1. Introduction: There is no mention in the introduction why the authors focused on the sexual lifestyle and covid-19.

We added the information P3 in 89-99 lines. Most of the COVID-19 studies conducted till today have focused on assessing the impact of the pandemic on physical health. This prompted us to reflect on the mental and sexual consequences that we consider to be equally important.

What is more, as I mentioned in our cover letter, my name is Anna Fuchs MD, PhD and I have a great pleasure to to conduct a research project on women's sexual health. My team my team consists of doctors guided by titled and experienced Professor Agnieszka Drosdzol- Cop.We are truly interested in the topic of gynaecology and obstetrics and for few years we have expanded our knowledge and research in this area. That is why we focused on the sexual lifestyle and COVID-19.

 #2. Methods: They commented on some data in this manuscript; however, if they would like to describe something about the results, they first need to perform multivariate analysis.

Thank you very much for your valuable remark. All data analyses were conducted using StatSoft Statistica version 13.0 PL software (Dell, TX, USA) and a p-value <0.05 was considered as significant. Analyses of dependent variables before and during quarantine were evaluated by the Wilcoxon’s rank test. For the sake of quantitative variables, comparison Chiˆ2 was utilized. Kruskal–Wallis test was implemented for independent variables, and further investigations were accomplished by the U Mann–Whitney test. Data are presented as a mean±SE (Standard Error).

The analysis did not find additional factors that could affect the average stress score among the respondents during the pandemic. This is our pilot study and we will certainly try to perform suggested by Reviewer multivariate analysis in future manuscripts.

#3. Figure 3 and 4: what the hell.

We sincerely apologize for technical difficulties in opening - Figure 3 and 4. Of course, we improved it.

#4. Discussion: The sentences are lengthy and include unnecessary sentences. and #5. P8, L212-226: The sentences are not needed.

Thank you very much for your valuable remark. We improved the discussion according to your comments and removed unnecessary paragraphs. Also, the manuscript has been reviewed by a native speaker.

#6. Discussion: I cannot understand what the authors like to conclude from the data.

The main finding of our study was an important influence of COVID-19 pandemic on the quality of sexual life and frequency of intercourse among polish women. We wanted to draw clinicians' attention to the very important role of sexuality in a woman's life, as the attention of doctors mainly focuses on infectious aspects, it would be beneficial to focus also on the long-term effects of a pandemic disrupting the quality of life.

Thank you once again for all your suggestions. We strongly believe that we have fulfilled your expectations because we put a lot of work into this manuscript.

Yours sincerely,

The Authors

Round 2

Reviewer 3 Report

The authors replied on each comment sincerely and the replies were appropriate. The quality of papers submitted for consideration includes enough reader's interest and scientific quality. The given paper satisfies requirements for publication of this journal.